# A novel index to assess low energy fracture risks in patients prescribed antiepileptic drugs

Ola Nordqvist[1,2]*, Olof Björneld[1,3,4], Lars Brudin[5,6], Pär Wanby[1,5,7], Rebecca Nobin[8,9], Martin Carlsson[1,10]

**1** Department of Medicine and Optometry, eHealth Institute, Linnaeus University, Kalmar, Sweden, **2** The Pharmaceutical Department, Region Kalmar County, Kalmar, Sweden, **3** Department of Computer science and Media technology, Data Intensive Sciences and Applications (DISA), Faculty of Technology, Linnaeus University, Kalmar, Sweden, **4** Business Intelligence, IT Division, Region Kalmar County, Kalmar, Sweden, **5** Department of Medical and Health Sciences, University of Linkoping, Linköping, Sweden, **6** Department of Clinical Physiology, Region Kalmar County, Kalmar, Sweden, **7** Section of Endocrinology, Department of Internal Medicine, Region Kalmar County, Kalmar, Sweden, **8** Department of Clinical Sciences, Lund University, Malmö, Sweden, **9** Department of Orthopedics, Region Kalmar County, Kalmar, Sweden, **10** Department of Clinical Chemistry, Region Kalmar County, Kalmar, Sweden

* ola.nordqvist@lnu.se

## Abstract

### Objective

To develop an index assessing the risks of low energy fractures (LEF) in patients prescribed antiepileptic drugs (AED) by exploring five previously suggested risk factors; age, gender, AED-type, epilepsy diagnosis and BMI.

### Methods

In a population-based retrospective open cohort study we used real world data from the Electronic Health Register (EHR) in Region Kalmar County, Sweden. 23 209 patients prescribed AEDs at any time from January 2008 to November 2018 and 23 281 matching controls were followed from first registration in the EHR until the first documented LEF, disenrollment (or death) or until the end of the study period, whichever came first. Risks of LEF measured as hazard rate ratios in relation to the suggested risk factors and in comparison to matched controls were analyzed using Cox regression. The index was developed using a linear combination of the statistically significant variables multiplied by the corresponding regression coefficients.

### Results

Data from 23 209 patients prescribed AEDs and 2084 documented LEFs during a follow-up time of more than 10 years resulted in the **K**almar **E**pilepsy **F**racture **R**isk **I**ndex (**KEFRI**). KEFRI = Age-category x (1.18) + Gender x (-0.51) + AED-type x (0.29) + Epilepsy diagnosis-category x (0.31) + BMI-category x (-0.35). All five previously suggested risk factors were confirmed. Women aged 75 years and older treated with an inducing AED against epilepsy and BMIs of 25 kg/m$^2$ or below had 48 times higher LEF rates compared to men aged

**Data Availability Statement:** These data were provided for analysis by Region Kalmar County in accordance with the approval from the ethical review board of Linköping University. Whilst the

data are pseudonymized, they are highly granular health data linked to individual health care clients in Kalmar County, crucially, no informed consent has been given for research use of these routine health data. For this reason the Region Kalmar County does not permit to open sharing but instead only grants primary use permission for the data: they are too sensitive to be openly shared. Other users would need to make the application themselves and each requested use case is reviewed by the Region Kalmar County. Data cannot be shared publicly because access to these data was only granted for primary use, and no permission was granted for secondary use. Re-use of this dataset requires approval from the Region Kalmar County. Please contact Professor Cecilia Fagerström (Cecilia.Fagerstrom@regionkalmar.se), Head of the Research Section at Region Kalmar County to request access to data extracted from the Electronic Health Records (Cambio COSMIC).

**Funding:** M.C. received funding from The Kamprad Family Foundation O.N reciveid funding from Region Kalmar County The funders had no role in study design, data collection and analysis, decision to publish, or preparation of the manuscript.

**Competing interests:** The authors have declared that no competing interests exist.

50 years or younger, treated with a non-inducing AED for a condition other than epilepsy and BMIs above 25 kg/m$^2$.

## Conclusion

The KEFRI is the first weighted multifactorial assessment tool estimating risks of LEF in patients prescribed AEDs and could serve as a feasible guide within clinical practice.

## Introduction

With an ageing population worldwide, the number of low energy fractures (LEF) is expected to increase from the 9 million annual fractures registered at the turn of the century [1]. These fractures cause both suffering and generate considerable health care costs [2]. Risk factors for LEFs have been assessed in several epidemiological studies [3, 4], resulting in the development of general risk assessment tools now being used in clinical practice. One of these tools is FRAX$^{®}$, which integrates eight clinical risk factors in estimating the 10-year major osteoporotic fracture risk [5]. Some of these risk factors, such as age, gender and BMI apply to the population in general. One of the more specific risk factors included in FRAX$^{®}$ is secondary osteoporosis, usually defined as low bone mass in the presence of an underlying disease or drug [6]. Drugs causing bone loss and thus increasing the risk of LEF have been stressed as an important area within drug safety [7]. The only drug class included in FRAX$^{®}$ at present is glucocorticoids.

Contradictory to this drug class selection, increased fracture rates in patients using antiepileptic drugs (AED) have been recognized ever since the 1960s [8] when risks were initially presented in institutionalized patients with epilepsy. The use of AEDs have since disseminated into other medical areas such as psychiatry [9] and pain medicine [10] resulting in a substantial increase in AED consumption. The fracture risk among AED users has been extensively investigated and is evidently multifactorial [11]. Drug side effects including impaired gate stability [12] and influence on bone mineral density [13] are considered contributing factors. The inducing effect on hepatic Cytochrome P450 (CYP) enzymes attributed to some AEDs, have been associated with vitamin D deficiency, and thus as one further underlying cause of both bone impairment and LEF, suggesting fracture risk differences between AED types [14]. In addition to side effects of AEDs, the medical conditions themselves can contribute to the fracture risk, e.g. due to seizure-related falls [14]. An epilepsy-specific risk assessment tool has been requested [15], since neither the epilepsy diagnosis nor use of AEDs are included in the FRAX$^{®}$ definition of secondary osteoporosis. In the recently updated version of QFracture$^{®}$, another osteoporotic fracture risk tool, epilepsy (either diagnosed or prescribed anticonvulsants) has been added, but merely as a single binary risk factor [16]. The multifactorial nature of the fracture risk among AED users calls for a more differentiated approach in risk assessment.

Data on drug prescriptions, medical diagnoses and events are registered in Electronic Health Records (EHR) together with basic demographic parameters. This type of real world data can act as a source in creating risk prediction models in health care [17]. In this study we developed a multifactorial risk assessment tool for LEFs in patients prescribed AEDs by applying a combination of five previously suggested risk factors; age, gender, BMI, AED-type and epilepsy diagnosis on data from EHR. We also compared the risk for LEFs in patients prescribed AEDs with a matched control group for risk factor confirmation.

## Methods

We conducted a retrospective, population-based, open cohort study. Our study followed the Strengthening the Reporting of Observational Studies in Epidemiology (STROBE) reporting guideline [18]. The regional ethical review board of Linköping University approved the study plan and deemed it exempt from informed consent because the data was pseudonymized.

### Data source

The EHR of Region Kalmar (Cambio COSMIC) was first introduced in 2008 and reached full coverage in 2010. The EHR covers the entire Kalmar County population, varying from 233 400 (in 2008) to 244 700 (in 2018). Relevant data regarding health events, prescriptions, vital parameters, clinical coding, and lab results are included in addition to basic demographic parameters. Both public and private care providers are included in primary as well as secondary care. Both somatic and psychiatric care is included in the EHR.

### Study cohort

Patients prescribed at least one AED during the period or having initiated treatment before January 2008 were initially included (Fig 1). For comparative outcome analysis a control group was selected with computer-based randomization based on gender and the age-decade interval at study initiation, ratio 1:1. The control group was defined as having no AED prescriptions before or during the study period. Patients were followed from first registry in the EHR and last follow-up date was defined as either decease date or date of the latest registration in EHR regardless of health condition. The study included data from January 2008 to November 2018 initially comprising 46 775 participants (AED patients = 23 396, and controls = 23 379). Later, 12 patients were excluded because of missing initiation dates (index dates) and 5 patients due to missing age information. Also, 268 patients were excluded because of fracture dates preceding the first AED prescription initiation date.

### Primary outcome

The primary outcome measure was defined as documented LEF i.e. a combination of ICD-10 codes according to the Swedish National Board of Health and Welfare's definition of osteoporosis-related fractures [19] and ICD-10 codes for low energy trauma (LET). Osteoporosis-related fractures were thus defined as ICD-10 codes: S22 (Fracture of rib(s), sternum and thoracic spine), S32.1—S32.8 (Fracture of lumbar spine and pelvis), S42.2 (Fracture of upper end of humerus), S42.3 (Fracture of shaft of humerus), S52.5 (Fracture of lower end of radius), S52.6 (Fracture of lower end of ulna), S72.0—S72.4 (Fracture of femur) and S82.1 (Fracture of upper end of tibia). LET was defined as ICD-10 codes for external causes of mortality and morbidity for low energy trauma by codes starting with: W00—W08 (Fall on same level). The registration of LET codes is compulsory in conjunction with a fracture registration in the EHR in Region Kalmar County. The fracture code and LET code had to be registered on the same healthcare event in order to be included.

### Variable definitions

For each participant demographic information on age, gender and date of decease were obtained. Furthermore, information on diagnosis, BMI, and drug prescription were extracted.

Age was calculated from date of birth to first registration date in EHR and age was categorized as: < 18, 18–50, 51–74 and ≥ 75 years.

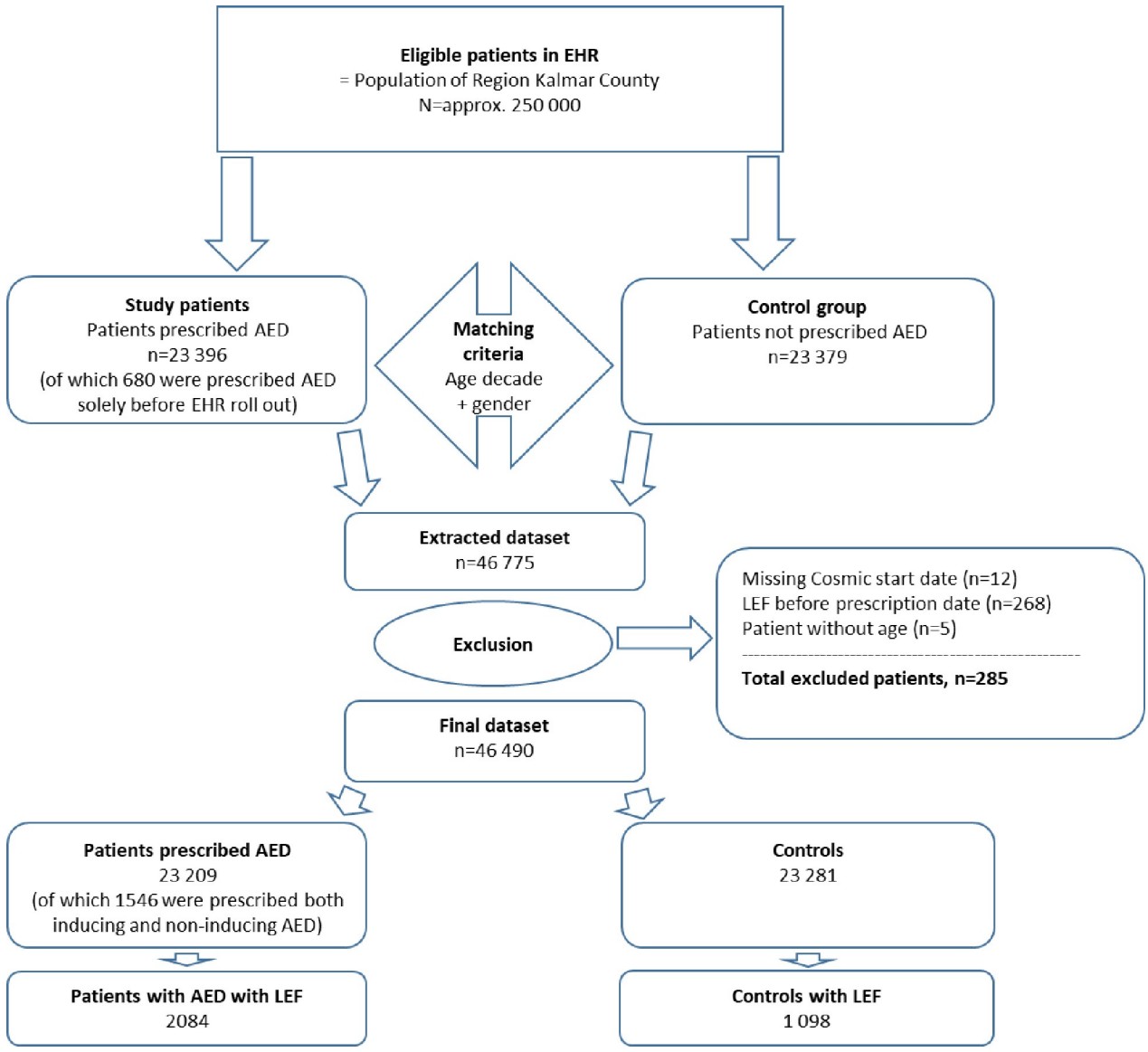

**Fig 1. Flowchart for patient and control group inclusion and primary outcome.**

Epilepsy was defined by ICD-10 codes starting with: G40 (Epilepsy and recurrent seizures), G41 (Status epilepticus) or F803 (Acquired aphasia-epilepsy syndrome). Diagnoses registered by medical secretaries and/or physicians were included regardless of level or type of care situation.

BMI was categorized as: $\leq 25$ and $> 25$ kg/m$^2$. The values used were either the registered BMI values or values calculated by registered measurements of length and weight.

Antiepileptic drugs were identified and classified into two groups based on their enzyme inhibiting profile. The group inducing CYP enzymes contained ATC codes: N03AA02 (phenobarbital), N03AA03 (primidone), N03AB02 (phenytoin), N03AB05 (fosphenytoin), N03AF01 (carbamazepine), N03AF02 (oxcarbazepine), N03AF03 (rufinamide) and N03AF04 (eslicarbazepine).

The group non-inducing CYP-enzymes contained ATC codes: N03AD01 (ethosuximide), N03AE01 (clonazepam), N03AG01 (valproate), N03AG04 (vigabatrin), N03AG06 (tiagabine), N03AX03 (sultiame), N03AX09 (lamotrigine), N03AX10 (felbamate), N03AX11 (topiramate),

N03AX12 (gabapentin), N03AX14 (levetiracetam), N03AX16 (pregabalin) and N03AX17 (stiripentol). Groups included in the statistical analyses were: inducing, non-inducing, and control group (no AED prescription)

Patients with one or more prescriptions for AED were included. Patients with only one type of AED during the study period were classified accordingly, while patients with both types of AED were classified according to drug exposure. Drug exposure was represented by the total number of prescription days, i.e. all prescription periods were calculated and summarized for the study follow-up period. Patients having prescriptions of both types of AEDs, were categorized in the group with the highest number of prescription days. For 78 patients the number of prescription days were exactly the same, 39 patients were then included by randomization to inducing, and 39 to non-inducing. 23 209 patients prescribed an AED (20497 non-inducing and 2712 inducing) and 23 281 matching controls were thus included for statistical analysis.

## Statistical analyses

We used a Cox univariate regression analysis followed by a multivariate analysis (p<0.1). For variables displaying statistical significance in the multivariate analysis, an index was developed for patients prescribed AED using a linear combination of the variables multiplied by the corresponding regression coefficients. Since the age groups < 18 and 18–50 years had identical fracture frequencies (1.7%) the two categories were lumped together in the final Cox regression giving the three categories $\leq 50$, 51–74 and $\geq 75$ years. The comparative results were presented in Kaplan Meier graphs. The software Statistica v.12 (StatSoft, Inc., Tulsa, OK, USA) was used for all analyses.

## Results

### Characteristics of patients and control group

Since the introduction of the EHR-system in Region Kalmar County in 2008, 23 209 patients having been prescribed AEDs in the EHR (20 497 with non-inducing AEDs + 2712 with inducing AEDs in Table 1) were included. This count also include the 680 patients which had AEDs prescribed solely prior to the EHR roll out, but having been registered in conjunction with the roll out (Fig 1). Of the 23 209 patients, 1546 patients had been prescribed both non-inducing and inducing AEDs during the study period (Fig 1). These patients were classified according to drug exposure i.e. the highest number of prescription days became the dominant AED type. The median age for participants' first registration in EHR was 55 years and nearly one of two patients prescribed AEDs were male (46.3%). A total of 3921 (16.8%) of the patients prescribed AEDs had an epilepsy diagnosis registered. The vast majority (20 497, 88.3%) of the patients prescribed AEDs, received drugs not considered as CYP enzyme inducers. The three most commonly prescribed non-inducing drugs were gabapentin, pregabalin and lamotrigine (82.5% of the non-inducing drugs prescribed) and the three most commonly prescribed enzyme-inducing drugs were carbamazepine, oxcarbazepine and phenytoin (90.8% of the inducing drugs prescribed). The age- and gender-matched control group consisted of 23 281 patients having a medical visit registered in the EHR, but having not been prescribed any AEDs during the period.

### Risk of low energy fractures in relation to previously suggested risk factors

In total, 2084 (1813 with non-inducing AEDs + 271 with inducing AEDs) patients (9.0%) among those prescribed AEDs suffered from LEFs (Table 1) during the median follow-up

**Table 1. Characteristics of patients prescribed antiepileptic drugs according to enzyme inhibiting profile and age- and gender-matched controls.**

| Variable | | Controls | AED-type Non-inducing | Inducing | Total |
|---|---|---|---|---|---|
| N | | 23281 | 20497 | 2712 | 46490 |
| **Gender** | | | | | |
| | Male (n; %) | 10758 (46.2) | 9129 (44.5) | 1612 (59.4) | 21499 (46.2) |
| | Female (n; %) | 12523 (53.8) | 11368 (55.5) | 1100 (40.6) | 24991 (53.8) |
| **Age (years)** | | | | | |
| | Mean (SD) | 52.0 (20.9) | 52.4 (20.6) | 50.5 (21.9) | 52.1 (20.8) |
| | Median (range) | 55 (0–108) | 55 (0–101) | 54 (0–100) | 55 (0–108) |
| **Age categories (years, n; %)** | | | | | |
| | <18 | 1697 (7) | 1383 (7) | 298 (11) | 3378 (7) |
| | 18–50 | 7980 (34) | 7147 (35) | 839 (31) | 15966 (34) |
| | 51–74 | 10315 (44) | 8942 (44) | 1236 (46) | 20493 (44) |
| | ≤75 | 3289 (14) | 3025 (15) | 339 (13) | 6653 (14) |
| **BMI (kg/m2)** | | | | | |
| | Mean (SD) | 27.5 (5.7) | 28.7 (6.3) | 27.8 (5.9) | 28.3 (6.1) |
| | Median (range) | 27 (13–82) | 28 (13–70) | 27 (14–61) | 27 (13–82) |
| | Missing (n; %) | 16188 (69.5) | 6783 (33.1) | 1331 (49.1) | 24302 (52.3) |
| **Alive at period end (n; %)** | | | | | |
| | Yes | 20985 (90.1) | 16260 (79.3) | 2042 (75.3) | 39287 (84.5) |
| | No | 2296 (9.9) | 4237 (20.7) | 670 (24.7) | 7203 (15.5) |
| **Epilepsy diagnosis (n; %)** | | | | | |
| | Yes | 46 (0.2) | 2804 (13.7) | 1117 (41.2) | 3967 (8.5) |
| | No | 23235 (99.8) | 17693 (86.3) | 1595 (58.8) | 42523 (91.5) |
| **Antiepileptic drugs (AED) (n; %)** | | | | | |
| | Gabapentin (N03AX12) | 0 (0.0) | 10004 (48.8) | 0 (0.0) | 10004 (21.5) |
| | Pregabalin (N03AX16) | 0 (0.0) | 4313 (21.0) | 0 (0.0) | 4313 (9.3) |
| | Lamotrigine (N03AX09) | 0 (0.0) | 2585 (12.6) | 0 (0.0) | 2585 (5.6) |
| | Valproate (N03AG01) | 0 (0.0) | 1676 (8.2) | 0 (0.0) | 1676 (3.6) |
| | Levetiracetam (N03AX14) | 0 (0.0) | 851 (4.2) | 0 (0.0) | 851 (1.8) |
| | Clonazepam (N03AE01) | 0 (0.0) | 782 (3.8) | 0 (0.0) | 782 (1.7) |
| | Topiramate (N03AX11) | 0 (0.0) | 267 (1.3) | 0 (0.0) | 267 (0.6) |
| | Vigabatrin (N03AG04) | 0 (0.0) | 11 (0.1) | 0 (0.0) | 11 (0.0) |
| | Ethosuximide (N03AD01) | 0 (0.0) | 7 (0.0) | 0 (0.0) | 7 (0.0) |
| | Sultiame (N03AX03) | 0 (0.0) | 1 (0.0) | 0 (0.0) | 1 (0.0) |
| | Carbamazepine (N03AF01) | 0 (0.0) | 0 (0.0) | 2173 (80.1) | 2173 (4.7) |
| | Oxcarbazepine (N03AF02) | 0 (0.0) | 0 (0.0) | 165 (6.1) | 165 (0.4) |
| | Phenytoin (N03AB02) | 0 (0.0) | 0 (0.0) | 125 (4.6) | 125 (0.3) |
| | Phenobarbital (N03AA02) | 0 (0.0) | 0 (0.0) | 118 (4.4) | 118 (0.3) |
| | Fosphenytoin (N03AB05) | 0 (0.0) | 0 (0.0) | 98 (3.6) | 98 (0.2) |
| | Primidone (N03AA03) | 0 (0.0) | 0 (0.0) | 27 (1.0) | 27 (0.1) |
| | Rufinamide (N03AF03) | 0 (0.0) | 0 (0.0) | 6 (0.2) | 6 (0.0) |
| **Long term epilepsy medication (≥5 yrs) (n; %)** | | | | | |
| | Yes | 0 (0.0) | 5609 (27.4) | 1248 (46.0) | 6857 (14.7) |
| | No | 23281 (100) | 14888 (73) | 1464 (54) | 39633 (85) |
| **LEF during follow up period (n; %)** | | | | | |
| | Yes | 1098 (4.7) | 1813 (8.8) | 271 (10.0) | 3182 (6.8) |

*(Continued)*

**Table 1.** (Continued)

| Variable | | Controls | AED-type Non-inducing | Inducing | Total |
|---|---|---|---|---|---|
| | No | 22183 (95) | 18684 (91) | 2441 (90) | 43308 (93) |

Age and age categories is the age when patients had the first registry in EHR.

period of more than 10 years (124.7 months, IQR 87.5 to 130.6 months) after the first registration in the EHR. The hazard rate ratios of LEFs were analyzed with regards to five previously suggested risk factors; age, gender, type of AED, epilepsy diagnosis and BMI (Table 2). In the univariate Cox regression analysis, the statistical differences in individual risk factors are displayed. Advanced age, female gender, having been prescribed an AED, having a registered epilepsy diagnosis and a low BMI all involved higher LEF rates. In the multivariate Cox regression analysis, the five risk factors' relative importance is presented, taking the variation of the risk factors between groups into account. All five risk factors remained statistically significant in the multivariate regression analysis. Long term medication ($\geq$ 5 years) was also incorporated in the model but was not found statistically significant, and did not alter the regression coefficients included in the index significantly. Patients prescribed CYP enzyme-inducing AEDs had a hazard rate ratio for LEF of 1.63 (1.38 to 1.92) compared to controls, while patients prescribed non-inducing AEDs had a ratio of 1.23 (1.13 to 1.35) compared to controls. Patients prescribed AEDs for epilepsy had a ratio for LEF of 1.40 (1.23 to 1.59) compared to those prescribed AED for non-epileptic conditions. In Figs 2 and 3, the differences in

**Table 2. Results from Cox proportional hazard regression regarding risks for low energy fractures, analysed for the total cohort.**

| Parameter | | N total | Fractures | Fractures (%) | Univariate Cox regression HR (95% conf. limits) | p | Multivariate Cox regression HR (95% conf. limits) | p |
|---|---|---|---|---|---|---|---|---|
| **Age categories (years, n)** | | | | | | | | |
| | **<18** | 3378 | 57 | 1.7 | **0.06 (0.05–0.08)** | **<0.001** | **0.03 (0.01–0.11)** | **<0.001** |
| | **18–50** | 15966 | 274 | 1.7 | **0.06 (0.05–0.07)** | **<0.001** | **0.07 (0.06–0.08)** | **<0.001** |
| | **51–74** | 20493 | 1552 | 7.6 | **0.28 (0.26–0.31)** | **<0.001** | **0.34 (0.31–0.37)** | **<0.001** |
| | **≥75** | 6653 | 1299 | 19.5 | 1.00 | | 1.00 | |
| **Gender (n)** | | | | | | | | |
| | **Female** | 24991 | 2210 | 8.8 | 1.00 | | 1.00 | |
| | **Male** | 21499 | 972 | 4.5 | **0.53 (0.49–0.57)** | **<0.001** | **0.57 (0.52–0.62)** | **<0.001** |
| **Group (n)** | | | | | | | | |
| | **Controls** | 23281 | 1098 | 4.7 | 1.00 | | 1.00 | |
| | **Non-inducing** | 20497 | 1813 | 8.8 | **1.57 (1.46–1.70)** | **<0.001** | **1.23 (1.13–1.35)** | **<0.001** |
| | **Inducing** | 2712 | 271 | 10.0 | **1.88 (1.65–2.15)** | **<0.001** | **1.63 (1.38–1.92)** | **<0.001** |
| **Epilepsy diagnosis (n)** | | | | | | | | |
| | **No** | 42523 | 2756 | 6.5 | 1.00 | | 1.00 | |
| | **Yes** | 3967 | 426 | 10.7 | **1.57 (1.42–1.74)** | **<0.001** | **1.40 (1.23–1.59)** | **<0.001** |
| **BMI (kg/m2, n)** | | | | | | | | |
| | **≤25** | 6999 | 997 | 14.2 | 1.00 | | 1.00 | |
| | **>25** | 15189 | 1444 | 9.5 | **0.61 (0.56–0.66)** | **<0.001** | **0.70 (0.64–0.75)** | **<0.001** |

Age category is the age for the patient's first entry in EHR. HR is hazard rate ratio. The multivariate analyses were reduced from 46490 to 22188 due to missing BMI values.

Significant results are presented in bold.

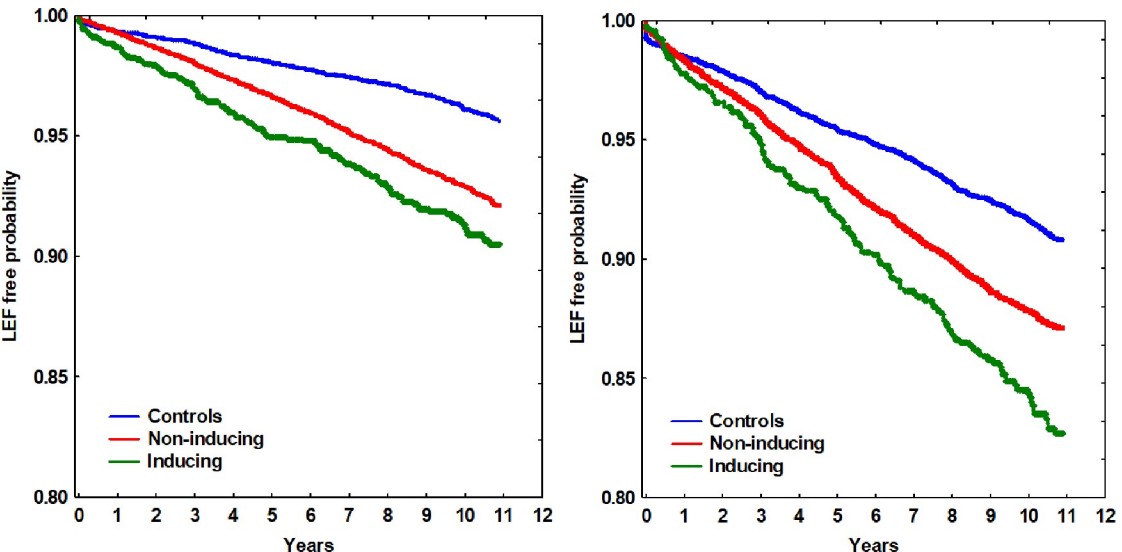

**Fig 2.** Low energy fracture-free probability over time in relation to type of antiepileptic drug, compared to controls for men (left) and women (right), respectively.

LEF risk associated with type of AED (Fig 2) and epilepsy diagnosis (Fig 3) compared to controls over time are shown in women and men, respectively.

## Visualization of combined risk factors

The differences in LEF risk when combining a number of risk factors are displayed in Fig 4. The group with the lowest fracture risk during the follow-up period in the study was used as reference (LEF risk 1.00) and risks in all other groups were calculated as hazard rate ratios in relation to the reference. Women aged 75 years and older treated with an inducing AED

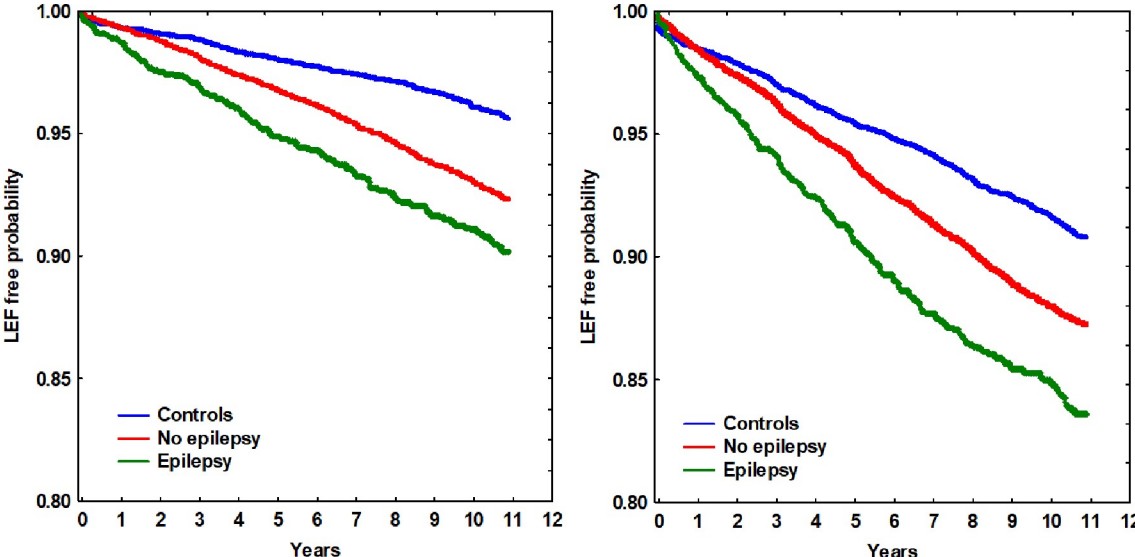

**Fig 3.** Low energy fracture-free probability over time in relation to being prescribed antiepileptic drugs and having a registered epilepsy diagnosis or not, compared to controls for men (left) and women (right), respectively.

| | Men | | | | Age | Women | | | | |
| | Non-epilepsy | | Epilepsy | | | Non-epilepsy | | Epilepsy | | |
| AED-type | High BMI | Low BMI | High BMI | Low BMI | Age | High BMI | Low BMI | High BMI | Low BMI | AED-type |
| Ind | 15 | 21 | 20 | 29 | ≥75 | 25 | 36 | 34 | 48 | Ind |
| Non-ind | 11 | 16 | 15 | 22 | | 19 | 27 | 26 | 36 | Non-ind |
| | | | | | | | | | | |
| Ind | 4.6 | 6.6 | 6.3 | 8.9 | 51-74 | 7.7 | 10.9 | 10.5 | 14.9 | Ind |
| Non-ind | 3.5 | 4.9 | 4.7 | 6.7 | | 5.8 | 8.2 | 7.8 | 11.1 | Non-ind |
| | | | | | | | | | | |
| Ind | 1.4 | 2.0 | 1.9 | 2.7 | ≤50 | 2.4 | 3.4 | 3.2 | 4.6 | Ind |
| Non-ind | 1.0 | 1.5 | 1.4 | 2.1 | | 1.8 | 2.5 | 2.4 | 3.4 | Non-ind |

Reference group is male, non-epilepsy, non-inducing AED type, youngest age group and high BMI.

**Fig 4. Chart displaying relative low energy fracture risks (as hazard rate ratios) during the follow-up period in relation to the five risk factors (gender, age, epilepsy diagnosis, type of antiepileptic drugs and BMI).** Age group is the age of the patient's first entry in the EHR.

against epilepsy and BMIs of 25 kg/m$^2$ or below thus had 48 times higher LEF rates compared to men aged 50 years and younger, treated with a non-inducing AED for a condition other than epilepsy, and BMIs above 25 kg/m$^2$ (the reference group).

## KEFRI: A tool to assess risk in clinical practice

Using the risk factors in the multiple regression analysis for patients prescribed AEDs in our study, the following equation was developed to act as a clinical risk assessment index. The weighted index is named the **K**almar **E**pilepsy **F**racture **R**isk **I**ndex (**KEFRI**).

**KEFRI** = Age-category x (1.18) + Gender x (-0.51) + AED-type x (0.29) + Epilepsy diagnosis-category x (0.31) + BMI-category x (-0.35)

| | | |
|---|---|---|
| Age categories (years): | ≤ 50 | =1 |
| | 51-74 | =2 |
| | ≥ 75 | =3 |
| Gender: | Women | =1 |
| | Men | =2 |
| AED type: | Non-inducing | =1 |
| | Inducing | =2 |
| Epilepsy diagnosis: | No | =1 |
| | Yes | =2 |
| BMI-category (kg/m$^2$): | ≤ 25 | =1 |
| | > 25 | =2 |

## KEFRI-stratification

In Fig 5, patients prescribed AEDs are stratified in quartiles according to the KEFRI index. Patients with an index of below 0.93 are the 25% having the lowest risk of LEFs during the follow-up period from first registry in the EHR. For these low risk patients, the 10-year risk of LEF is approximately 3%. Patients with a KEFRI index of 0.93 to 1.59 have an 8% risk, whereas patients with an index of 1.60 to 2.06 have a 10-year risk of 18%. The quartile with the highest 10-year LEF risk are those with a KEFRI index of above 2.06. For these patients, the risk is 27%.

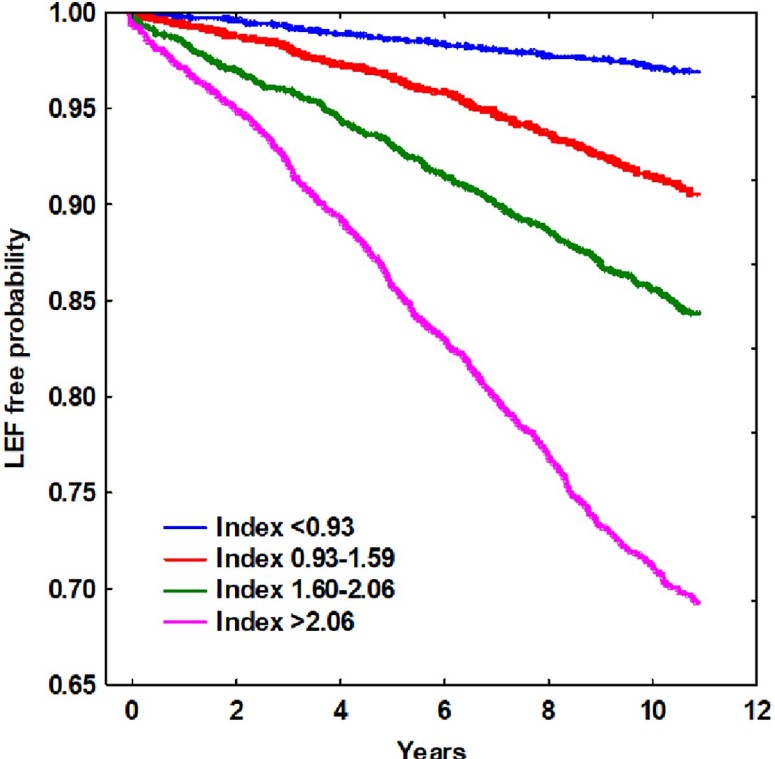

**Fig 5. Low energy fracture-free probability over time in relation to KEFRI-value (quartiles).**

## Discussion

There is a growing interest for generating clinical evidence using real world data from EHR. This retrospective open cohort study used such real world data to confirm previously suggested risk factors, but also used the weighted importance of these risk factors in order to create an index for assessing risk of LEFs in patients using AEDs. We found that both the type of AED and the combination with an epilepsy diagnosis are indeed important factors, supporting the claim that it is not enough to consider AED-treatment or epilepsy as a single binary risk factor, as in the updated QFracture®, let alone altogether overlook it, as in FRAX®.

Several meta-analyses have been published on the subject of fractures in epilepsy/use of AEDs [20, 21]. These reviews have assessed various studies on risks and associations in different cohorts.

Shen et al. [11] published the most recent systematic review of patients using AEDs regardless of diagnosis in 2014. Fifteen studies were included for analysis. In this meta-analysis the heterogeneity-adjusted relative risk for all fractures was found to be 1.85 (1.62 to 2.12) among AED users globally. If taking only cohort studies into account (9 studies) it was 1.97 (1.31 to 2.96). If including only osteoporosis-related fractures (3 studies) the relative risk was 1.88 (1.40 to 2.53), and among the studies only including patients >50 years (13 studies) it was 1.85 (1.61 to 2.13). When comparing AED types (inducing and non-inducing) to controls (4 and 4 studies, respectively) the differences in fracture risks were 1.60 (1.26 to 2.02) and 1.27 (1.02 to 1.59), respectively. The difference between inducing and non-inducing (4 studies) was 1.18 (1.11 to 1.25). In our study, 20 497 patients with non-inducing and 2712 with inducing AEDs were included. Overall, the patients with AEDs had a doubled unadjusted relative risk of LEFs, and we observed a 23% (13 to 35%) risk increase of LEF in patients with non-inducing AEDs,

and a 63% (38 to 92%) risk increase in patients with inducing AEDs compared to controls (in the multivariate analysis) which is well in line with the meta-analysis from Shen et al.

Risks of fractures in patients with epilepsy using AEDs have been specifically reviewed [21]. The three studies included in the review showed mixed results. Two studies (one [22] classified as poor quality and one [23] of good quality) suggested no risk difference between inducing and non-inducing, while the other (classified in the review as having high quality) study found a 22% (12 to 34%) risk difference between AED types among women and a borderline difference (9% (-2 to 20%)) among men [24]. Two of the studies had relatively short follow-ups (about 2 years) and the third study follow-up time was unclear. The study having good quality which showed no AED-type differences did however find that the risk of fracturing increased with cumulative AED exposure among patients with epilepsy [23]. This is also indicated in our study where the mean follow-up time was longer than 10 years. We observed a relative risk of LEF increase of 40% (23 to 59%) among patients with epilepsy in our multivariate analysis. This is a lower risk increase compared to that of the meta-analysis by Vestergaard [14] where 5 studies of measuring any kind of fracture were included, which stated a relative risk of 2.2 (1.9 to 2.5). Our lower risk increase could be explained by the fact that we only included LEF. It is previously known that about a third of the fractures in patients with epilepsy are linked to seizures [14].

Risks in different age groups have been specifically studied. These studies indicate that risks of fractures increase with (most) AED use in children [25], adults and the elderly [26, 27]. In our study 3364 older people with AEDs of 6653 (75 years or older) were included. We observed an increased relative fracture risk of 66% (63 to 69%) in the elderly compared to those middle-aged (51–74 years). In our study there were 1681 with AEDs of 3378 children included. Very few LEF (N = 57) were documented during a follow-up of more than 10 years. This is probably due to the fact that most fractures in children are results of high energy/trauma.

Since women generally have higher risks of bone impairment, osteoporosis and fractures, fracture risk in postmenopausal women with AEDs have been assessed [28] showing a hazard ratio of 1.44 (1.29 to 1.65) compared to matched women controls without AED treatment. Gender differences in fracture risk among epilepsy patients have also been studied [23] where women have had higher fracture rates in relation to treatment duration. In our study there was a near doubling of risk of LEFs among women compared to men (57% (52 to 62%) of the risk in women in the multivariate analysis).

Although the association between Body Mass Index and fracture risk is complex [29] BMI below 25 kg/m$^2$ is considered a risk factor for major osteoporotic fractures, when unadjusted to BMD. This was confirmed in our study even though we were only able to retrieve BMI values from half of the patients (48%) from the EHR. There was a 30% (25 to 36%) lower risk of LEF in patients with a BMI of >25kg/m$^2$.

Our study has several limitations that should be addressed. First, our control groups were matched only by age and gender. This could possibly result in confounding effects due to differences in socioeconomic factors, hereditary risks and comorbidity. Second, we included all LEFs and AEDs and did not stratify the results in relation to fracture location or individual AED. Furthermore we did neither take multiple AED use nor AED dosage into account. Third, the few patients prescribed osteoporosis treatment (less than 2% in the population) were not excluded. We would have included further risk factors e.g. cigarette smoking and alcohol intake in our model, however in the current EHR there was no standardized way of recording these variables; thus, they were left out. KEFRI is based on merely five previously suggested risk factors and gives a crude estimation of LEF risks over time. While other risk assessment tools have integrated treatment duration in the index or have estimated risk in relation to a predetermined period (e.g. 10-year risk) [5], we found the aspect of temporal associations to be of a complex nature. Patients in our study were included from first registry in EHR

and followed until first documented LEF, disenrollment (or death) or until the end of the study period, whichever came first. When including long term treatment (total AED prescription $\geq$ 5 years) as a risk factor, we found that it did not have a significant influence on KEFRI. The drug exposure (total days of AED prescription) was therefore used to determine the AED type categorization only among patients prescribed multiple types of AED in our study. This implies that patients could have undergone treatment breaks during the follow-up period, this in turn influencing the LEF risk. Furthermore, treatment regimens could differ substantially between diagnoses when prescribing AED perplexing the risk estimation. In spite of the limitations this study is strengthened by the fact that the entire population of patients in a Swedish region prescribed AEDs, regardless of diagnosis, were included, that the follow-up time was longer than 10 years, that we were able to look at LEFs registered by physicians rather than all fracture types, and that the multivariate analysis allowed us to use the weighted risk factors to develop a fracture risk index among the patients in the Kalmar region prescribed AEDs. To determine KEFRIs performance, the algorithm needs to be tested in an external population henceforth.

A recent review by Miziak [30] provided an update on the issue of drugs causing LEFs. Several publications stress investigation with bone mineral density (BMD) and subsequent treatment in patients using AEDs [20, 31, 32]. Despite this, awareness of the risks seem to be low among neurologists and in primary care [33], resulting in the patient group being overlooked [34]. Furthermore, the lack of fracture risk awareness seems to be low among patients with AEDs, whereby only 30% are aware of the association [35].

We think that the KEFRI could act as a future tool in clinical practice to increase the awareness among physicians regarding risks and risk factors in this specific patient group. KEFRI could possibly be integrated in the EHR as a decision support system, when assessing risks of LEFs during consultations. Furthermore KEFRI could create awareness among individual patients using AEDs through physician-patient dialogue.

## Author Contributions

**Conceptualization:** Ola Nordqvist, Olof Björneld, Lars Brudin, Pär Wanby, Rebecca Nobin, Martin Carlsson.

**Data curation:** Olof Björneld.

**Formal analysis:** Ola Nordqvist, Lars Brudin.

**Investigation:** Ola Nordqvist.

**Methodology:** Ola Nordqvist, Olof Björneld, Lars Brudin, Pär Wanby, Rebecca Nobin, Martin Carlsson.

**Supervision:** Ola Nordqvist, Martin Carlsson.

**Validation:** Ola Nordqvist, Olof Björneld, Lars Brudin.

**Visualization:** Ola Nordqvist, Pär Wanby.

**Writing – original draft:** Ola Nordqvist, Martin Carlsson.

**Writing – review & editing:** Ola Nordqvist, Olof Björneld, Lars Brudin, Pär Wanby, Rebecca Nobin, Martin Carlsson.

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
