## [Decision Letter · Decision Letter 0]

19 Jan 2021

PONE-D-20-40318

A novel index to assess low energy fracture risks in patients prescribed antiepileptic drugs- Using real world data from Swedish EHR to create the KEFRI: Kalmar Epilepsy Fracture Risk Index

PLOS ONE

Dear Dr. Nordqvist,

Thank you for submitting your manuscript to PLOS ONE. After careful consideration, we feel that it has merit but does not fully meet PLOS ONE’s publication criteria as it currently stands. Therefore, we invite you to submit a revised version of the manuscript that addresses the points raised during the review process.

The reviewers have noted some serious, overlapping concerns.  Ultimately, they converge on the model and coding exposures, covariates, and outcomes.  The revision must address these issues, and ideally provide a more complete accounting of the logic underlying the choices you've made.

We look forward to receiving your revised manuscript.

Kind regards,

Robert Daniel Blank, MD, PhD

Academic Editor

PLOS ONE

Journal Requirements:

Reviewers' comments:

Reviewer's Responses to Questions

**Comments to the Author**

1. Is the manuscript technically sound, and do the data support the conclusions?

Reviewer #1: Partly

Reviewer #2: Yes

2. Has the statistical analysis been performed appropriately and rigorously? 

Reviewer #1: No

Reviewer #2: Yes

3. Have the authors made all data underlying the findings in their manuscript fully available?

Reviewer #1: Yes

Reviewer #2: Yes

4. Is the manuscript presented in an intelligible fashion and written in standard English?

Reviewer #1: Yes

Reviewer #2: Yes

5. Review Comments to the Author

Reviewer #1: This study aimed to develop a fracture prediction tool for patients with epilepsy. There are several major limitations that need to be addressed prior to publication:

- The study population was chosen using a 1:1 matching design of AED users and controls by age and gender. All these factors: age, gender and AED are also risk factors used for the development of KEFRI. Such a design is not ideal for the development of a risk prediction model because the distribution of risk factors is regulated by design and almost certainly does not represent the real distribution of these risk factors in the population at risk, thus affecting generalisability and external validation.

- KEFRI index, was developed from a Cox Model but does not have a time specific baseline risk in the equation.

- There are no risk factors specific to epilepsy: number of seizures, falls

Specific comments:

How was epilepsy diagnosed? Is clinical diagnosis derived from hospitalisations, or emergency department presentations or GP/specialist consultations or all? It is not clear why people with an epilepsy diagnosis were included among controls. Please justify.

Fracture: what happened to MVA fractures or high trauma fracture? Were they excluded as event or as participant? Such details are needed in the methodology.

Variable definition:

- Suggest developing the model based on continuous rather than categorical variables, as they produce more accurate results than the categorical. If not, strong reasons should be provided for the use of categorical data.

Drug exposure:

1. How many prescriptions/days covered are needed to be included in the drug category?

2. The membership to a specific AED type needs to be explained in more details. The methodology just state that the highest number of prescriptions was used as criteria to membership to a specific AED.

More detail is needed:

How many people were classified to the second prescribed AED type?

Was there any wash-out period considered between the stop and start of new AED type?

Can it be assumed that past use of a certain AED type is equal to no use? Please explain if any consideration was given to past use of AED and how the drug intervals were created in participants who transitioned from an AED type to another.

Statistical analysis section is very brief and leaves out needed details:

- How was time of follow-up constructed? What is the start and stop of follow-up times?

- Did the authors use a paired Cox proportional hazards model to account for the case-control design?

- How was the accuracy and predictive power of the model examined?

Results

- The presence of only 16% epilepsy diagnosis among the cases seems to be a strong limitation for a study aiming to develop an index risk for fracture in this population.

- Table 1: need a p-value for the significance between each AED type medication vs controls.

- Fracture risk:

o Two of the fracture risk factors: age and gender were used for matching between AED and controls. How did this study design allow for the assessment of independent effect of age, gender and AED on fracture risk?

- Table 2: I would suggest choosing the group with lowest fracture risk as control in the analysis (i.e. male gender and BMI≤25). It will also be reasonable to collapse age into only 2 groups (younger and older than 50).

- Age and BMI should also be analysed as continuous variables with HR presented as an increment per SD.

- Replace comma with dot in the percentage column

- Table 3: A legend is needed for the colour code used to differentiate the magnitude of fracture risk according to risk factors. Please revise the use of comma for separate integer and decimal places.

- KEFRI equation needs to be revised. A baseline risk (i.e. 1-year, or 5-year) function needs to be added to all predictive models derived from Cox models. Otherwise, the equation will provide the same fracture estimates regardless of the time to fracture.

- Figure 1 needs to be edited. It is not clear who are the eligible participants in box 1 from which the dataset is extracted. The algorithm of matching AED and controls also need to be specified in this first step. I would recommend collapsing all three exclusions steps into one which will state all the exclusion criteria. A last box should indicate the number of fractures for each group.

- For Figures 2, 3, 4 the correct terminology is fracture-free probability and not proportion non-fracture. Both label and caption need to be revised for all figures.

Reviewer #2: This study uses electronic health record data from one geographic region of Sweden to develop a simple index to estimate relative risk of fractures in individuals prescribed anti-epileptic drugs (AED). The strengths of the study are that the study cohort is broad and population based, the study data tease apart the contributions of AED type and diagnosis (indication for AED) to fracture risk, and the index is simple and easily calculated from common EHR variables. A very good rationale for the study was presented in the Introduction.

There are important weaknesses and limitations that in my view need to be addressed before this is accepted for publication, detailed below.

Major Concerns

1. Important covariates were not considered for the model. Most important, prior fracture (before start of index period Jan 2008) and smoking were left out of the model. These are important confounders, since they are both associated with primary exposure variables (AED and/or epilepsy diagnosis) and outcome variable (incident fracture).

2. I am concerned about the potential for misclassification of incident fracture status, since a low energy code (W00 to W08) was also required. At least in some parts of the world, physicians are not required to indicate level of trauma in their documentation of diagnosis. If this is also true in Kalmar, it is possible that in this study many with incident fractures were misclassified as not having had a fracture. This would not be much of a concern if physicians in Kalmar are required in their documentation to indicate level of trauma for the fracture, e.g. that this is a “hard stop” that prevents the physician from completing their documentation until this step is completed. This concern would also be mollified if there is validation data available in Kalmar regarding the accuracy of these diagnosis codes.

3. I am also concerned about potential misclassification of exposure. The first sentence of the Study Cohort section indicates that those having initiated an AED before January 2008 were included. If they had discontinued the AED before the start of the study period, were they still included in the exposed group, excluded, or included in the control group?

4. If individuals discontinued AED treatment during the follow-up period, were they censored, or assigned follow-up time in the control group? This needs to be clarified in the Statistical Analysis part of the Methods.

Minor Issues

5. Absolute fracture risk estimated by FRAX is part of national Swedish guidelines for identifying candidates for osteoporosis. Figure 4 is the potentially really valuable data from this study for application of this index within the context of Swedish guidelines. I suggest the authors consider showing a supplemental table of 10-year absolute risks in subsets defined by age group, BMI, AED type, and diagnosis (similar to table 3 but showing absolute 10 year risks rather than relative risks).

6. What proportion of Region Kalmar residents are not included in this dataset? I presume that with a single-payer system this is covering the entire population and none are missed, but it would be helpful for the authors to confirm that.

7. Primary outcome section page 5; please state the skeletal sites that correspond to these fracture codes (can add site in parenthesis after each code)

8. Variable definitions page 5; please add generic drug name after each ATC code.

9. Last row table 1 page 8; indicate these are incident fractures over follow-up period, not percent with prior fracture at baseline.

10. Related to Major Concern #2; please state what level of trauma these LET codes indicate (fall from standing height or less?)

11. Define long-term medication near bottom of page 8 (you state this is ≥ 5 years in the table but it should be restated here)

12. Second to last sentence bottom of page 8, I presume the comparator group are those prescribed AED but for non-epileptic conditions? Would state that for clarity.

13. For Table 3 page 10, define reference group in a footnote (no AED, youngest age group, male, BMI <=25).

6. PLOS authors have the option to publish the peer review history of their article (what does this mean?). If published, this will include your full peer review and any attached files.

Reviewer #1: No

Reviewer #2: No

---

## [Author Response · Author response to Decision Letter 0]

18 Mar 2021

Please find our response and revisions enclosed.

---

## [Decision Letter · Decision Letter 1]

12 Apr 2021

PONE-D-20-40318R1

A novel index to assess low energy fracture risks in patients prescribed antiepileptic drugs- Using real world data from Swedish EHR to create the KEFRI: Kalmar Epilepsy Fracture Risk Index

PLOS ONE

Dear Dr. Nordqvist,

Thank you for submitting your manuscript to PLOS ONE. After careful consideration, we feel that it has merit but does not fully meet PLOS ONE’s publication criteria as it currently stands. Therefore, we invite you to submit a revised version of the manuscript that addresses the points raised during the review process.

Please see editor and reviewer comments.

We look forward to receiving your revised manuscript.

Kind regards,

Robert Daniel Blank, MD, PhD

Academic Editor

PLOS ONE

Additional Editor Comments (if provided):

I agree with the reviewer that the time element must be addressed. There are 2 options, as I see it:

1. revise the proposed equation to incorporate time to event

or

2. make no change to the equation but address the point prominently in the discussion.

Ultimately, it is up to readers to decide what is useful and what is not. The role of the journal is to present the readers with sufficient information about the methods so they can make informed judgments. Transparency is the key, and I will insist on that.

Reviewers' comments:

Reviewer's Responses to Questions

**Comments to the Author**

1. If the authors have adequately addressed your comments raised in a previous round of review and you feel that this manuscript is now acceptable for publication, you may indicate that here to bypass the “Comments to the Author” section, enter your conflict of interest statement in the “Confidential to Editor” section, and submit your "Accept" recommendation.

Reviewer #1: All comments have been addressed

2. Is the manuscript technically sound, and do the data support the conclusions?

Reviewer #1: No

3. Has the statistical analysis been performed appropriately and rigorously? 

Reviewer #1: No

4. Have the authors made all data underlying the findings in their manuscript fully available?

Reviewer #1: Yes

5. Is the manuscript presented in an intelligible fashion and written in standard English?

Reviewer #1: Yes

6. Review Comments to the Author

Reviewer #1: I thank the authors for addressing my comments. The paper is clearer and improved.

However, there is still one point in which I disagree with the authors. I have raised the issue of using time in the KEFRI equation. The authors replied that the risk is constant over time, and this is indeed true when considering the relative risk (ie the risk of AED use vs control). However, Cox model calculates the risk within a certain time frame given the assumption that the person survived up to that point. Thus, an equation derived from Cox Model in which time is ignored, as in KEFRI, would make no difference between people who fracture at 2-year compared to those who fracture at 10-year, despite the latter having a much lower absolute risk.

Indeed, I am not aware of any risk calculator derived from a Cox model that does not have a time function in the equation. Here are a few examples of nomograms predicting absolute risks a 5 or 10-year risk1,2.

Given this considerations, I believe that in the current form KEFRI does not have a great clinical utility.

There are also a couple of minor issues:

Results –Time of follow-up should be in person-years with IQR.

Table 1: bold the significant differences between medication groups and controls

Legend 4- is confusing. I believe that the HR presented is compared to the lowest fracture risk group (men, younger than 50 and low BMI). Is this correct? A clear description of how the HR were calculated needed to be added for clarity.

References

1. Nguyen ND, Frost SA, Center JR, Eisman JA, Nguyen TV. Development of prognostic nomograms for individualizing 5-year and 10-year fracture risks. Osteoporos Int. Oct 2008;19(10):1431-1444.

2. Yang H-I, Sherman M, Su J, et al. Nomograms for Risk of Hepatocellular Carcinoma in Patients With Chronic Hepatitis B Virus Infection. Journal of Clinical Oncology. 2010;28(14):2437-2444.

7. PLOS authors have the option to publish the peer review history of their article (what does this mean?). If published, this will include your full peer review and any attached files.

Reviewer #1: No

---

## [Author Response · Author response to Decision Letter 1]

8 Jun 2021

Please find our responses to suggestions from editor and reviewer enclosed in the rebuttal letter

---

## [Decision Letter · Decision Letter 2]

2 Aug 2021

A novel index to assess low energy fracture risks in patients prescribed antiepileptic drugs- Using real world data from Swedish EHR to create the KEFRI: Kalmar Epilepsy Fracture Risk Index

PONE-D-20-40318R2

Dear Dr. Nordqvist,

We’re pleased to inform you that your manuscript has been judged scientifically suitable for publication and will be formally accepted for publication once it meets all outstanding technical requirements.

Kind regards,

Robert Daniel Blank, MD, PhD

Academic Editor

PLOS ONE

Additional Editor Comments (optional):

Reviewers' comments:

Reviewer's Responses to Questions

**Comments to the Author**

1. If the authors have adequately addressed your comments raised in a previous round of review and you feel that this manuscript is now acceptable for publication, you may indicate that here to bypass the “Comments to the Author” section, enter your conflict of interest statement in the “Confidential to Editor” section, and submit your "Accept" recommendation.

Reviewer #1: All comments have been addressed

2. Is the manuscript technically sound, and do the data support the conclusions?

Reviewer #1: Yes

3. Has the statistical analysis been performed appropriately and rigorously? 

Reviewer #1: Yes

4. Have the authors made all data underlying the findings in their manuscript fully available?

Reviewer #1: Yes

5. Is the manuscript presented in an intelligible fashion and written in standard English?

Reviewer #1: Yes

6. Review Comments to the Author

Reviewer #1: I thank the author for addressing all my comments. I am satisfied with authors's decision to address the temporal association in the discussion.

7. PLOS authors have the option to publish the peer review history of their article (what does this mean?). If published, this will include your full peer review and any attached files.

Reviewer #1: No

---

## [Editor Report · Acceptance letter]

18 Aug 2021

PONE-D-20-40318R2 

A novel index to assess low energy fracture risks in patients prescribed antiepileptic drugs 

Dear Dr. Nordqvist:

I'm pleased to inform you that your manuscript has been deemed suitable for publication in PLOS ONE. Congratulations! Your manuscript is now with our production department. 

Kind regards, 

on behalf of

Professor Robert Daniel Blank 

Academic Editor

PLOS ONE